# Community support for injured patients: A scoping review and narrative synthesis

Rashi Jhunjhunwala[1,2]*, Anusha Jayaram[1,2], Carol Mita[3], Justine Davies[4,5‡]*, Kathryn Chu[5‡]

1 Program in Global Surgery and Social Change, Harvard Medical School, Boston, Massachusetts, United States of America, 2 Department of Surgery, Beth Israel Deaconess Medical Center, Boston, Massachusetts, United States of America, 3 Countway Library, Harvard Medical School, Boston, Massachusetts, United States of America, 4 Institute of Applied Health Research, University of Birmingham, Birmingham, United Kingdom, 5 Centre for Global Surgery, Department of Global Health, Faculty of Medicine and Health Sciences, Stellenbosch University, Stellenbosch, South Africa

‡ JD and KC share joint last authorship on this work.
* J.Davies.6@bham.ac.uk (JD); rjhunjhu@bidmc.harvard.edu (RJ)

**Data Availability Statement:** All relevant data are within the manuscript and its Supporting Information files.

**Funding:** This study was conducted as part of the Equi-Injury Study, funded by the UK National

## Abstract

### Background

Community-based peer support (CBPS) groups have been effective in facilitating access to and retention in the healthcare system for patients with HIV/AIDS, cancer, diabetes, and other communicable and non-communicable diseases. Given the high incidence of morbidity that results from traumatic injuries, and the barriers to reaching and accessing care for injured patients, community-based support groups may prove to be similarly effective in this population.

### Objectives

The objective of this review is to identify the extent and impact of CBPS for injured patients.

### Eligibility

We included primary research on studies that evaluated peer-support groups that were solely based in the community. Hospital-based or healthcare-professional led groups were excluded.

### Evidence

Sources were identified from a systematic search of Medline / PubMed, CINAHL, and Web of Science Core Collection.

### Charting methods

We utilized a narrative synthesis approach to data analysis.

### Results

4,989 references were retrieved; 25 were included in final data extraction. There was a variety of methodologies represented and the groups included patients with spinal cord injury (N

Institute of Health and Care Research, NIHR Global Health Groups Call 3, application 133135. Authors who received award are JD and KC. The funders had no role in study design, data collection and analysis, decision to publish, or preparation of the manuscript.

**Competing interests:** The authors have declared that no competing interests exist.

= 2), traumatic brain or head injury (N = 7), burns (N = 4), intimate partner violence (IPV) (N = 5), mixed injuries (N = 5), torture (N = 1), and brachial plexus injury (N = 1). Multiple benefits were reported by support group participants; categorized as social, emotional, logistical, or educational benefits.

## Conclusions

Community-based peer support groups can provide education, community, and may have implications for retention in care for injured patients.

## Introduction

The World Health Organization (WHO) estimates that 4.4 million people die from injuries such as acts of violence, road traffic accidents, falls, and burns annually, and 90% of these deaths occur in low- and middle-income countries (LMICs) [1]. Millions more suffer non-fatal injuries that require extensive medical care and support [2]. Injured patients often face many barriers to seeking, reaching, and receiving care [3]. These barriers can prevent or delay rehabilitation that allows for return to optimal function, especially after the acute injury has been addressed. Given the potentially chronic conditions that result from injury, retention in care is necessary. However, injured peoples' needs go beyond retention in medical care; they can also require care for ongoing psychological issues or matters of daily living which are often not addressed within healthcare systems.

For patients with other medical conditions, peer and community support has been shown to facilitate access to healthcare through improved retention of current care, introduction to other providers of care, and provision of information and psychological support [4–7]. Peer support has long been used within mental health services and has been defined as social emotional support that is mutually offered or provided by those with similar lived experiences [8]. This process of support, companionship, and assistance often counter feelings of loneliness, discrimination, and frustration. The most common form of peer support has been self-help groups, which have been defined as a voluntary small group for mutual aid [9]. Studies on peer support amongst HIV patients also demonstrate increased retention in care, improved anti-retroviral therapy adherence and viral suppression, and increased financial and moral support [10, 11]. Further, studies of pregnant women demonstrate the positive impact that peer-support can have on motherhood and coping with issues like substance use [12, 13]. Engaging the community, especially for historically marginalized groups, has also been demonstrated to result in positive health outcomes [14, 15]. Further, meaningful engagement of and advocacy by members of the HIV/AIDS community has been shown to lead to improved HIV/AIDS services and policy changes that lead to better access to care and service provision [16, 17].

Despite the similar needs of injured patients for ongoing medical care and the often life-changing nature of their injuries requiring psychological and physical support, there is little known about whether community-based peer support (CBPS) groups exist, benefit injured persons, or whether they play a role improving and retaining access to acute and chronic injury care. This scoping review aims to identify the extent, distribution, benefits, utility, and impact of community-based peer support groups for injured patients.

## Methods

This scoping review is conducted based on the expanded Arksey and O'Malley framework and uses PRISMA-ScR guidelines [18, 19].

## Defining the research question

We aimed to answer the question: what research has been done on the extent, distribution, benefits, utility, and impact of health-system independent community-based support groups (i.e. those which are not based in healthcare facilities or run by healthcare professionals) for injured persons?

## Search strategy

Studies reporting on CBPS groups for physical injuries were identified by a systematic search of Medline / PubMed (National Library of Medicine, NCBI); CINAHL (CINAHL Complete, EBSCOhost), and Web of Science Core Collection (Clarivate). Controlled vocabulary terms (i.e., MeSH; CINAHL thesaurus subject headings) were included when available and appropriate. The search strategies were designed and carried out by a librarian (CM). No language limits or date restrictions were applied. The exact search terms used for each of the databases are provided in the S1 File.

These searches were undertaken on March 1st, 2023. We also searched the grey literature via Google searches, reviewed injury society and trauma society web pages, and communicated with global experts in the field of injury research to identify additional studies evaluating the impact of these groups.

## Study inclusion and exclusion

Study inclusion and exclusion criteria are shown in **Table 1**. We included primary research studies as well as those which describe advocacy or policy changes to manage injuries, since community support groups have been advocates for improved care for other conditions such as HIV/AIDS [16, 17].

We excluded studies that evaluated groups that were hospital or rehabilitation facility-based or led/created by healthcare providers. We also excluded all studies that focused on Post Traumatic Stress Disorder (PTSD) where it was not clear that the PTSD occurred because of physical injury.

## Study screening & data extraction

All studies found were uploaded into Covidence for screening and data extraction. Covidence is a web-based collaboration software platform that streamlines the production of systematic and other literature reviews [20]. Two reviewers (RJ, AJ) screened all titles and abstracts independently for inclusion. In case of disagreement, RJ and AJ resolved conflicts through discussion. This process was repeated for full text review in Covidence.

**Table 1. Inclusion and exclusion criteria.**

| Inclusion | Exclusion |
|---|---|
| • All publications through March 1, 2023<br>• Studies of community or patient groups focused on peer-support of people who have been injured and evaluated the impact of community or patient groups for those people<br>• Studies that assess the needs and priorities of community-based patient groups for people who have been injured<br>• All countries<br>• English language | • Studies that evaluated groups that were either led or created by hospital-affiliated staff, healthcare providers, or rehabilitation facilities<br>• Studies of community support groups that focused on injury prevention, self-harm, health issues not related to the injury, or groups that focused only on mental health or Post Traumatic Stress Disorder<br>• Support groups that did not specify that the participants had experienced physical injury |

A data extraction form was developed and each of two reviewers independently extracted data from each article, after which consensus was reached through review and discussion. Study data points collected were study design, funding, injury type, the study population, number of participants, what content was shared in the group, and discussion themes in the group. We also collected data on the country in which the support group was based, the country of the study's first and last author, and the years the study was conducted and published.

## Data analysis

We approached the data analysis for this review via a narrative synthesis methodology. This methodology was chosen as it is a widely utilized and accepted mechanism for conducting scoping reviews, as it is a way to synthesize and aggregate the existing body of knowledge on a topic to then elucidate areas of opportunity for further investigation. Since our inclusion criteria did not preclude any specific study methodologies, we expected variety and heterogeneity in our final study sample. We did not assess study quality in this scoping review as we believed all relevant articles per the inclusion criteria should be assessed.

## Synthesis of results

The outcomes were grouped into three categories. The first category was study characteristics, which includes study methodology, years during which the studies were conducted, geographic location of support groups, and presence or absence of study funding. We next reported on support group characteristics, including type of injury, number of participants, leadership, and facilitation of the groups, and whether the group itself received funding. Finally, we reported the summary of benefits described by the support group members–this was categorised by outcomes for all injury types together and then by specific injury type.

# Results

## Study characteristics

4,989 references (PubMed: 2343; CINAHL: 1077; Web of Science: 1569) were retrieved from the database searches on March 1, 2023. Duplicate records were removed using EndNote; import into Covidence resulted in 3489 unique references for screening. Of these, 112 were included for full text review. Full text was not recoverable for 13 studies due to unavailability online through Harvard libraries, inter-library loan, or other formal retrieval mechanisms. Four of 13 unavailable studies were published prior to 1990 and contact information for the authors was not available. Attempts to contact the other nine authors were unsuccessful. Of the 99 full texts that were reviewed, 25 studies met criteria for data extraction (**Fig 1**).

## Study characteristics

Of the 25 studies included, eleven used purely qualitative methodology [21–31], six were cross-sectional quantitative studies [32–37], three used mixed-methods [38–40], three were case reports [41–43], and there were two randomized controlled trials [44, 45]. Studies were published from 1981 to 2022. Eleven studies were conducted in the US [23, 25, 30, 31, 34, 36, 38, 40, 42, 43, 45], six in Canada [21, 22, 27–29, 44], and three in both countries [32, 35, 39]. There was one study each from the UK [37] and South Africa [41]. There were two studies in which the peer support occurred in the form of online forums and thereby were not limited by geographic region [24, 26]. **Table 2** presents an overview of the 25 studies included in data extraction.

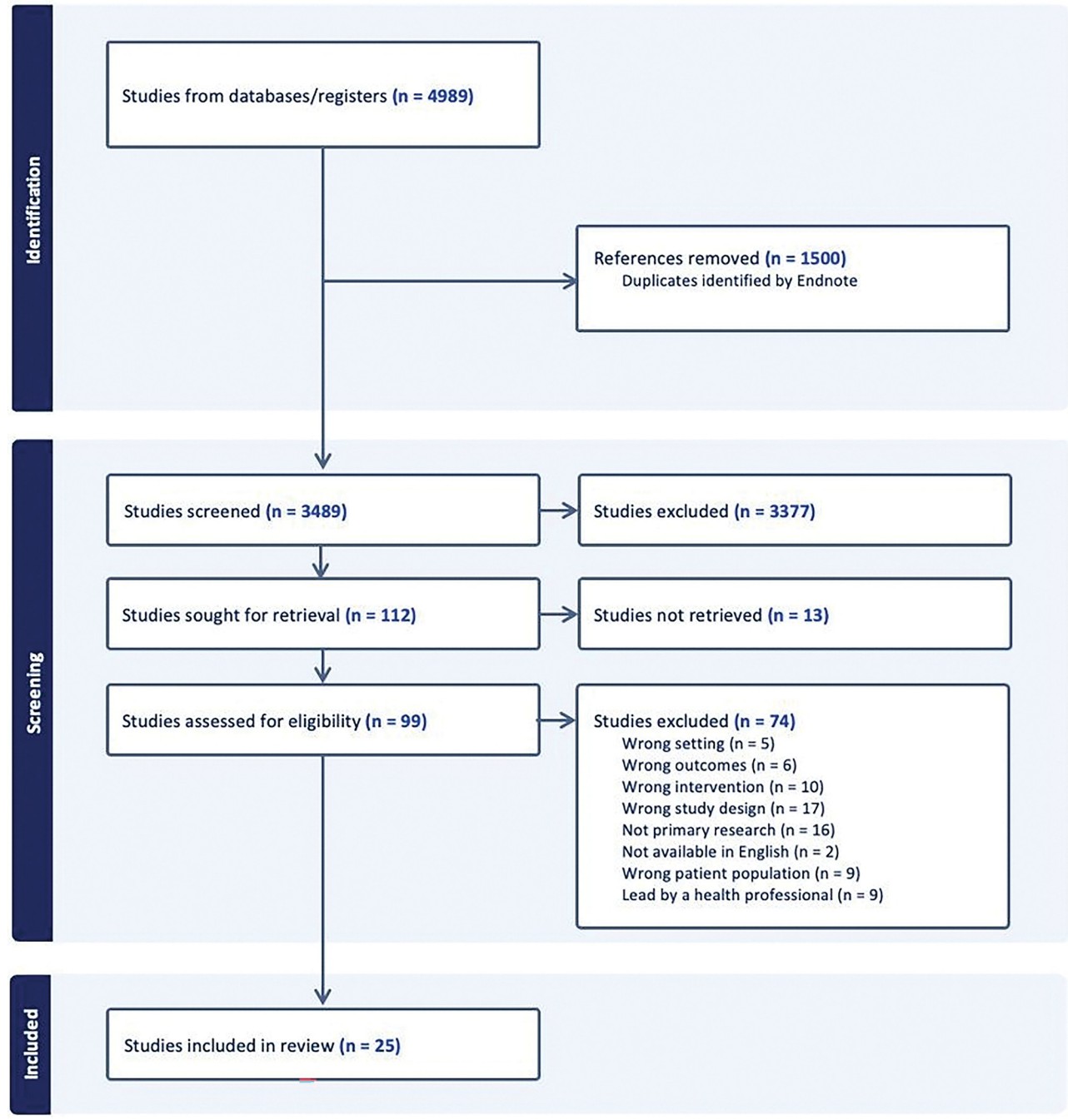

**Fig 1. PRISMA flow diagram.**

## Support group characteristics

CBPS groups have been studied for persons with spinal cord injury (N = 2), traumatic brain or head injury (N = 7), burns (N = 4), intimate partner violence (IPV) (N = 5), mixed injuries (N = 5), torture (N = 1), and brachial plexus injury (N = 1). The number of support group members ranged from 2 to 60+, with most of the groups reporting 20–40 active members. Fifteen studies reported on whether group leadership was provided by members themselves or an

**Table 2. Summary table of included studies.**

| Title | First Author, Year | Country in which the study was conducted | Data collection period | Funding for Group | Study Aim | Study Design | Injury Type | Number of participants | Study Findings |
|---|---|---|---|---|---|---|---|---|---|
| Comparison of a Cognitive-Behavioral Coping Skills Group to a Peer Support Group in a Brain Injury Population | Backhaus, 2016 | USA | Not Reported | None | To examine whether group CBT would result in improved perceived self-efficacy (PSE), emotional functioning, and neurobehavioral functioning compared with a structurally similar peer support group program for BI | Randomized control trial | Brain Injury | 38 | Individuals with BI and their caregivers may benefit from either group CBT or a peer support group, with a trend for those received group CBT to maintain the effects of improved PSE. For our review, we extracted data solely from the control group, which functioned as a support group in its own right. |
| Helping others heal: burn survivors and peer support | Badger, 2010 | US/Canada | Not Reported | None | To evaluate perceptions of peer support and the value they placed on it related to the key burn recovery outcome variables of: social comfort, life satisfaction, QOL (body image, affect, interpersonal relationships, sexuality, productivity, and social integration. | Cross Sectional | Burn | 95 | Burn survivors held a positive regard for peer support, with it being ranked as the best resource of the Phoenix society. |
| Adult burn survivors' views of peer support: a qualitative study | Badger, 2010 | Canada | 2007 | None | To learn from burn survivors whether peer support was important to them in their recovery and why, and secondarily, if they had volunteered as a peer supporter, if there were any negative consequences or advice they would have for others. | Qualitative | Burn | 30 | Burn survivors highly valued peer support, especially by providing hope, combating isolation, and lessening the feeling of being stigmatized due to physical scars. |
| Processes and outcomes of a communalization of trauma approach: Vets & Friends community-based support groups | Balmer, 2021 | USA | 2017 | None | To inform training and group leader support procedures to ensure the maximal functioning and effectiveness of V&F support groups | Qualitative | Mixed Injury group | 23 | Outcomes included restoration of trust, connection with group members, building skills to manage trauma, and community acceptance and engagement. It also found that Vets and Friends could possible meeting veteran specific needs. |
| Making meaning in a burn peer support group: qualitative analysis of attendee interviews | Davis, 2014 | USA | 2010–2011 | None | To better understand the psychosocial recovery needs of burn survivors and the role of peer support groups, by performing a series of guided qualitative interviews. | Qualitative | Burn | 6 interviewees, 6 key informants, 30–50 group members | The support group was instrumental to their psychosocial adjustment and growth. It allowed them to make meaning of their injury, learn coping strategies, make meaningful relationships, and have hope. |
| Constructing robust selves after brain injury: positive identity work among members of a female self-help group | Gelech, 2019 | Canada | 2012 | Saskatchewan Association for the Rehabilitation of the Brain Injured Shoppers Drug Mart Foundation for Women | To understand the work around identity in rehabilitation contexts by analysing how survivors construct the self in a female acute BI survivor self-help group. | Qualitative | Brain Injury | 5 | Four modes of identity work were seen: constructing competent selves, tempering the threat of loss and impairment, resisting infantilization and delegitimization, and asserting a collective gender identity. |
| Long-Term Social Reintegration Outcomes for Burn Survivors With and Without Peer Support Attendance: A Life Impact Burn Recovery Evaluation (LIBRE) Study | Grieve, 2020 | US/Canada | October 2014–December 2015 | None | To determine the association of peer group attendance with the societal reintegration of a large group of burn survivors from the US and Canada using the Life Impact Burn Recovery Evaluation (LIBRE) profile. | Cross Sectional | Burn | 601 responded to survey; 330 participated in peer support | Burn survivors who attended peer support groups had better scores on social iterations, social activities, and work and employment than those who did not. Those who attended also had larger burns and were more likely to be more than 10 years out from injury. |
| Accessing peers' and health care experts' wisdom: a telephone peer support program for women with SCI living in rural and remote areas | Jalovcic, 2009 | Canada | 2004 | None | To capture the essence of women's experiences of the Telephone Peer Support Group program and its main characteristics and structure as perceived by participants. | Qualitative | Spinal Cord Injury | 7 | The participatory approach of this group allowed participants to control the direction of the group, allowing for emotional support and exchange. There were 17 core themes that were grouped into four structures: space, relation to self, relation to others, and causality. |
| Group therapy for refugees and torture survivors: treatment model innovations | Kira, 2012 | USA | 2006–2008 | None | To develop group therapy models based on thorough and in-depth analyses of the nature of torture and its psychosocial effects through the study of a social organization for Bhutanese torture survivors. | Case Report | Torture/Refugee | 20 women (first group); 8–12 families (2nd group) | Group therapy for torture survivors is an important part of the comprehensive rehabilitation approach for survivors, and that group therapies extend into community healing. |

(Continued)

**Table 2.** (Continued)

| Title | First Author, Year | Country in which the study was conducted | Data collection period | Funding for Group | Study Aim | Study Design | Injury Type | Number of participants | Study Findings |
|---|---|---|---|---|---|---|---|---|---|
| Observations from practice: support group membership as a process of social capital formation among female survivors of domestic violence | Larance, 2004 | USA | January 2000–December 2002 | Jersey Battered Women's Service, Inc.'s Community Outreach Program | To explore the authors' practice observations of female domestic violence survivors' journey from first agency contact to active participation in a support group process. | Mixed Methods (Participant Observation and Survey Data) | Physical Intimate Partner Violence | 412 | Support groups provide abused women with the opportunity to build trust and networks, with demonstrations of calling on these relationships outside of the agency's confines for support and enjoyment. |
| Using an integrated knowledge translation approach to inform a pilot feasibility randomized controlled trial on peer support for individuals with traumatic brain injury: A qualitative descriptive study | Lau, 2021 | Canada | 2021 | None | (1) To understand key informants' perspectives of the barriers and facilitators of participating in peer support research and programs among individuals with TBI; (2) to understand key informants' perspectives on the perceived impacts of peer support programs on individuals with TBI; and, (3) to demonstrate how an iKT approach can inform the development and implementation of a pilot feasibility randomized controlled trial (RCT). | Qualitative | Brain Injury | 22 | There were five main themes related to the barriers and facilitators to participating in peer support research and programs: knowledge, awareness, and communication; logistics of participating; readiness and motivation to participate; need for clear expectations; and matching. There were three main themes related to the perceived impact of peer support: acceptance, community, social experiences; vicarious experience/learning through others: shared experiences, role-modelling, encouragement; and "I feel better." |
| A Pilot Feasibility Randomized Controlled Trial on the Ontario Brain Injury Association Peer Support Program | Levy, 2021 | Canada | 2021 | None | The overall objective of this study was to determine the feasibility of conducting an RCT and the preliminary effectiveness of the OBIA Peer Support Program | Randomized control trial | Brain Injury | 13 | The study found no statistically significant results for community integration, mood, or self-efficacy; however, changes in these outcomes were accompanied by moderate-to-large effect sizes. Within health-related quality of life, the mean pain score of the intervention group was significantly lower than that of the control group at the two-month timepoint but not at completion. Interviews revealed proximal improvements in knowledge, skills, and goals, and identified two domains related to trial acceptability: (1) environmental context and resources, and (2) reinforcement. |
| Unexpected barriers in return to work: lessons learned from injured worker peer support groups | MacEachen, 2007 | Canada | 2004 | None | To study injured worker peer support groups for an empirical picture of return-to-work difficulties, including how peer support performs both positive and negative return-to-work functions at the individual and policy level. | Qualitative | Mixed injury group | 37 | The study identified four dimensions of peer support: worker experience of being misunderstood by system providers, need for advocates, social support, and help with procedural complexities of the workers' compensation and health care systems. It concluded that peer support groups can be part of the return to work for workers, but there is also a need for more sensitivity to structural and social issues. |
| Support groups for injured workers: process and outcomes | Mignone, 1999 | Canada | 1992–1994 | Workers' Compensation Board of Alberta | This study evaluated the impact of support groups on injured workers with musculoskeletal injuries in relation to pain, somatization, depression, and pain-locus-of-control. | Cross Sectional | Mixed Injury group | 62 in intervention; 40 comparison | The study found that between those who did and didn't participate in the support groups, there was no demonstrable effect on the well-being of those who did participate, but it could not rule out any benefits of participating in support groups. |
| Divorcing abused Latina immigrant women's experiences with domestic violence support groups | Molina, 2009 | USA | Not Reported | None | The purpose of this study was to explore and describe the experiences of Latina immigrants who had been members of 1 year-long domestic violence support groups. | Cross Sectional | Physical Intimate Partner Violence | 15 | Overall the group experience was positive, with members learning about the domestic violence cycle, how to understand their children, their childhood abuse, and legal information on their situation. The members felt like they made friends, felt protected, comfortable, and understood, and gave them courage to fight. |

(*Continued*)

**Table 2.** (Continued)

| Title | First Author, Year | Country in which the study was conducted | Data collection period | Funding for Group | Study Aim | Study Design | Injury Type | Number of participants | Study Findings |
|---|---|---|---|---|---|---|---|---|---|
| From isolation to connection: understanding a support group for Hispanic women living with gender-based violence in Houston, Texas | Morales, 2009 | USA | 2005 | La Rosa Family Services | To examine how the community based organization's support group encourages immigrant Hispanic women to focus on developing tools and strategies, self-esteem, and empowerment. | Mixed Methods (Archival research, Interviews, & Participant Observation) | Physical Intimate Partner Violence | 30 | Women enjoyed the benefits of the support group and were taught to manage situations, emotions, and becoming less dependent on the abuser. They also became more aware of their self-worth and value and are making efforts towards self-sufficiency. |
| A Thematic Analysis of Online Discussion Boards for Brachial Plexus Injury | Morris, 2016 | Internet | 2015 | None | The purpose of this study was to examine posts on Internet discussion groups dedicated to brachial plexus injury and identify common ideas through thematic analysis | Qualitative | Brachial Plexus Injury | 336 posts | Themes focused on emotional aspects of brachial plexus injury and information support related to the disease, treatment, recovery after treatment, and process of seeking care. |
| Helping the patient cope with the sequelae of trauma through the self-help group-approach | Scanlon-Schlipp, 1981 | USA | Not Reported | None | To describe the development of Trauma Recovery, a self-help group | Case Report | Mixed injury group | n/a | The self-help group is an effective means for recoverees to engage in therapeutic activities and is cost efficient. The Trauma Recovery program has a psychosocial emphasis and has established a groundwork for regional chapters. |
| Helping factors in a peer-developed support group for persons with head injury, Part 2: Survivor interview perspective | Schulz, 1994 | USA | 1993 | Vital Active Life after Trauma Inc | The purpose of the study was to determine the perspective of group members relative to helping factors in their peer-developed support group | Qualitative | Brain Injury | 4 | Eleven helping factors were generated: socializing, finding out other people's perspectives and attitudes, expressing thoughts and feelings, receiving support, feeling something in common, gaining understanding, empathy and acceptance through listening and sharing, getting perspective, helping others, getting help, feeling hope, and learning more about BI. |
| Helping factors in a peer-developed support group for persons with head injury, Part 1: Participant observer perspective | Schwartzberg, 1994 | USA | Not Reported | Vital Active Life after Trauma Inc | An ethnographic study to identify the helping factors in a peer-developed support group for persons who survived a traumatic head injury. | Qualitative | Brain Injury | 13 | Positive attributes in a successful peer support group were feeling a part of the group, having a common problem and being able to validate the effects of an injury, and receiving information through a variety of channels. |
| Promoting group empowerment and self-reliance through participatory research: a case study of people with physical disability | Stewart, 1999 | South Africa | Not Reported | None | The purpose of this case study of a SCI self-help group is (1) To share the experiences of both the health professional and group members in moving from an originally professionally-led group to an independent peer-led group (2) To demonstrate how participatory research may serve as a tool for group empowerment and self-reliance; and (3) To illustrate how a critically informed professional group work praxis may enable marginalized groups to take action for social change. | Case Report | Spinal Cord Injury | 12 | This case study shows how group participation with a participatory research framework can unlock self-reliance and empowerment. It also offers insight to the group process and the role of health professionals to create opportunities for empowerment and self-reliance for people with a disability. |
| The Effect of a Trauma Risk Management (TRiM) Program on Stigma and Barriers to Help-Seeking in the Police | Watson, 2018 | UK | November 2012–January 2013 | None | The primary aim of this study was to investigate differences between Trauma Risk Management (TRiM) and non-TRiM-using forces on help-seeking and stigma among serving police personnel. In addition, the study explored whether TriM had any effect on PTSD symptoms. | Cross Sectional | Mixed Injury group | 859 | The TriM group reported lower levels of PTSD symptoms and showed less stigmatized views towards mental health difficulties. They also percieved fewer barriers to help-seeking. |

(*Continued*)

**Table 2.** (Continued)

| Title | First Author, Year | Country in which the study was conducted | Data collection period | Funding for Group | Study Aim | Study Design | Injury Type | Number of participants | Study Findings |
|---|---|---|---|---|---|---|---|---|---|
| Online Conversations About Abuse: Responses to IPV Survivors from Support Communities | Whiting, 2022 | Internet | 2019 | None | In this analysis we sought to discover the types of responses being offered in reply to survivors' self-disclosures and questions regarding their experiences with IPV. | Qualitative | Physical Intimate Partner Violence | 451 | There were three main categories of responses from the data, including support, sharing experiences, and sharing information. The subcategories included specific types of help, advice, understanding, and were overwhelmingly positive in their tone and content. |
| The self-help movement in head injury | Williams, 1990 | USA | Not Reported | None | A survey to describe self-help groups and their benefits | Cross Sectional | Brain Injury | 77 | The main focus of meetings was social support. Membership varies by time after injury, sex, and relationship to survivors, and these variables impact how the support group meets the needs of its members. Group leaders should often survey the group to ensure needs are being met. The study suggests that nurses may provide an important role in recruitment, information, and group facilitation. |
| Support groups for older victims of domestic violence | Wolf, 2001 | US/Canada | 1997 | Domestic violence programs, ageing services | To identify support groups for older victims of abuse, to gather information that would be useful in assisting communities to meet the needs of these elders, and to determine whether the type of sponsorship (aging services or domestic violence programs) made a difference in group operations. | Mixed Methods (cross sectional survey & interviews) | Physical Intimate Partner Violence | 30 support groups | Recommendations for future groups include accessibility of the meeting site, using a leader familiar with aging issues or an elder themselves, resources for recruitment, and a steady source of funding. Resistance of elders to participate was noted to be a large barrier. |

external facilitator such as professional facilitators, therapists, social workers, or social work or therapy students [22, 25, 27, 28, 31, 34, 36–38, 40–45]. Of these fifteen studies, seven [22, 25, 36–38, 40, 42] reported that the groups were run by professionals rather than participant volunteers, two groups were run by non-professional facilitators [31, 34]. Five groups reported that peers volunteered or rotated through leading the groups [28, 41, 43–45] and one group utilized one-to-one peer mentorship [27]. Data about the group facilitators were unavailable or unclear for the remaining studies. Seven peer-support groups reported receiving funding for the groups' activities or meetings–one from the Alberta Worker's Compensation Board [33] and six from philanthropic organizations [23, 25, 28, 38–40].

## Benefits reported by support group participants

There were a variety of positive benefits reported by support group members. These were grouped into educational benefits, social benefits, emotional benefits, and logistical benefits (Fig 2). Educational benefits include explanations and descriptions of various disease processes and what participants could expect for their disease course. Social benefits were defined as those benefits that ameliorated the participants' social network, solidarity with others with similar lived experiences, and community building. Emotional benefits are those benefits that participants indicated were impactful to their sense of self or emotional safety and wellbeing. Lastly, logistical benefits are those that directly pertain to making connections with aid

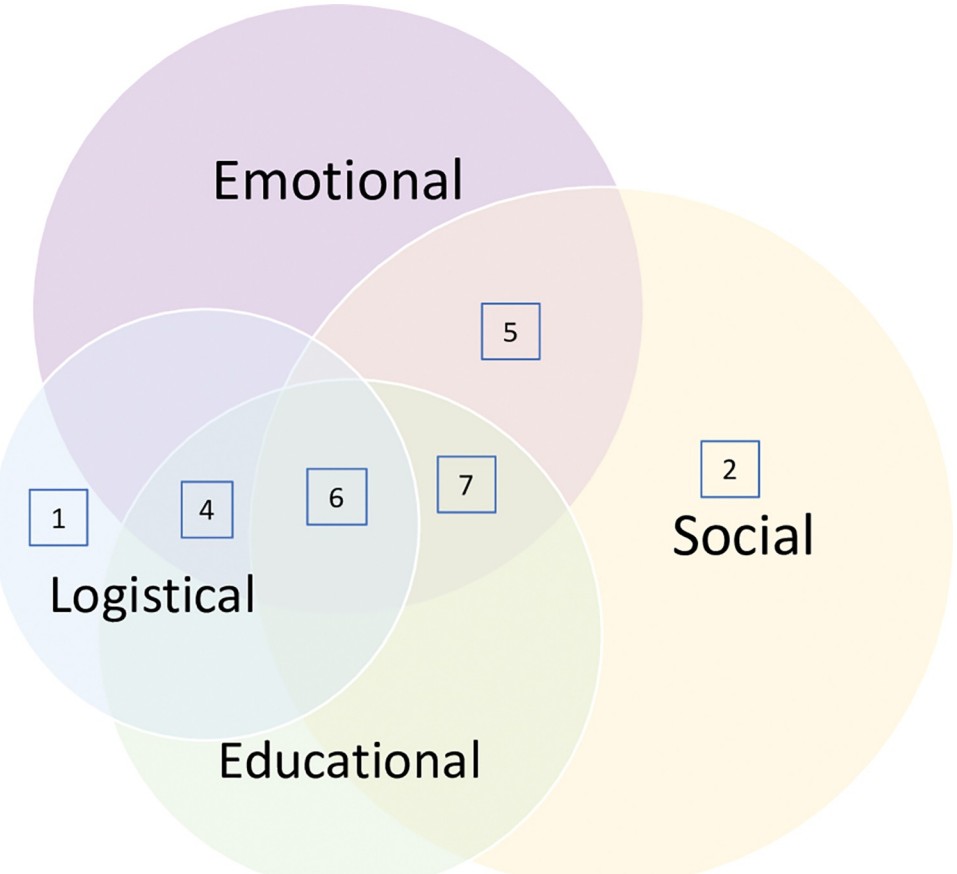

**Fig 2. Categories of peer support reported in all studies; number of studies per grouping indicated within boxes.**

Table 3. Peer support type reported by injury group.

| Injury type | Total # of Groups | Emotional support | Educational support | Social support | Logistical support |
|---|---|---|---|---|---|
| Brain injury | 7 | 7 | 7 | 7 | 4 |
| Burn injury | 4 | 3 | 1 | 4 | 0 |
| IPV | 5 | 4 | 1 | 5 | 2 |
| Mixed injury | 5 | 4 | 2 | 4 | 4 |

services, legal assistance, guidance on workers' compensation claims, and other administrative issues.

Thirteen of 25 studies concluded that shared educational or didactic material in the group was a benefit of participation [21–27, 32–34, 41, 44, 45]. Eight studies reported that they valued and appreciated the sharing of medical knowledge, either via dissemination from the group leadership or between participants within the groups [22–25, 28, 33, 34, 44].

From a social perspective, nineteen studies reported that support group members indicated they were able to obtain guidance and coping strategies as a result of group participation [21–23, 25–27, 29–31, 33–36, 38–45], and fourteen studies reported that the groups created a sense of solidarity and group identity that members found lacking in other areas of their lives [22, 23, 27–31, 34, 38–43]. Social interaction was mentioned as a positive outcome in eleven studies [21–23, 27, 31, 32, 35, 36, 40, 42, 44].

Emotional outcomes constituted a substantial portion of the reported benefits. Fourteen of the studies reported that participants found emotional support and/or catharsis as a function of group membership [22–27, 31, 33, 36, 38, 42–45], and fourteen studies reported that group members felt a sense of not feeling alone [21, 22, 25–33, 36, 40, 43] after they joined the group. Group participants in eleven studies reported a benefit to their self-esteem, empowerment, and overall quality of life [22, 28–32, 36, 38, 41, 42, 44]. Six studies noted that support groups gave participants a sense of hope for the future [23, 29–31, 43, 44]. Finally, seven studies reported that participants felt a sense of altruism and being able to give back to other injury survivors as a result of group participation [23, 25, 29, 30, 34, 41, 43]. On a logistical level, eleven studies reported that group members found that they were able to gain assistance in navigating the health, legal, and/or social systems from their peers [21, 23–25, 31, 33, 34, 36–38, 44].

We also evaluated the types of benefits reported by groups for people with specific types of injuries. We focused on studies reporting on brain injury support groups, burn injury support groups, intimate partner violence (IPV) support groups, and mixed injury support groups, as these injury types were the subject of more than two studies included in this review. Table 3 shows the distribution of each type of support reported by each injury group.

Considering the types of benefits reported by people with specific types of injuries, out of the seven studies reporting on brain injury support groups, six studies mentioned that participants guidance and coping strategies from the groups [23, 25, 27, 34, 44, 45]. Of the four studies reporting on persons with burn injuries, three reported that participants appreciated the feeling of camaraderie and reduced isolation [29, 30, 32]. All five studies reporting on intimate partner violence support groups reported that participants gained a sense of group identity or solidarity [26, 36, 38–40]. Lastly, four of the five support groups for mixed injuries reported that they appreciated peer assistance in navigating health and legal systems [21, 31, 33, 37].

## Discussion

We found 25 studies which assessed the benefits of CBPS groups for injured persons, globally. The majority were created for specific types of injury, including spinal cord, brain or head,

and brachial plexus injuries as well as burns. There were also groups for intimate partner vio-lence (IPV), torture, and some for mixed injuries. All injury support groups except one were located in high-income countries. Most commonly, participants noted gaining social support, emotional support and catharsis, and educational or didactic materials from CBPS groups. Mental health support was the main reported benefit with few data on improved physical health or retention in care.

Our review identified positive benefits of CBPS for injured people, including group identity, regaining of self-esteem, emotional support and guidance, delivery of practical knowledge about their disease, recovery and treatment process, or ways to access medical, legal, or social support. This mirrors the positive effects of support groups for people living with diabetes [46], mental health problems [47], and heart disease [48], which have been shown to dissemi-nate practical knowledge and health education, social and emotional support, navigation of the medical system, and building trust-based relationships. In addition, the psychological com-ponents of both intentional [49] (i.e. interpersonal violence) as well as unintentional (road traf-fic crashes) injuries can be substantial [50]. Further, and highly applicable to injured patients, peer-led and community-based support groups have implications for rehabilitation, which can reduce limitations in functionality [51] especially in LMICs when access to medical care can be difficult to obtain. CBPS groups have been shown to be useful for retention in care and both physical and mental health improvement in other conditions, and thus offer potential for peo-ple who have been injured.

People with injuries potentially face unique challenges in finding peer support compared to people seeking support for specific disease processes, since trauma can result in an array of injuries even given the same mechanism [52]. While we found four mixed-injury support groups, twenty-one were injury-specific. Thus, CBPS groups for injured persons are difficult to set up because persons with injuries may not identify with just one injury type. Further-more, we did not find support groups for people with certain injury types. For example, there were no studies that reported on CBPS groups for abdominal or thoracic trauma, which can result in significant disability and often is followed by fragmented care in which patients suffer unplanned admissions often at multiple different care sites. This fragmentation ultimately iso-lates patients and contributes to worse outcomes [53]. Additionally, there were no specific CBPS groups for people who have been injured by certain mechanisms, such as road traffic injuries or firearm injury, although these mechanisms are some of the highest contributors to death and disability worldwide [54–56]. Victims of these types of trauma might have similar health needs having gone through the same traumatic mechanism, and peer-support could be useful in navigating their care post-injury. These broader groups can also act as powerful advo-cacy agents; by bringing people with various injuries and mechanisms together, there is a greater opportunity to advocate for general injury prevention and access to consistent and unfragmented care [57–59].

We found only one study that was conducted in an LMIC setting. Given the high burden of injury in LMICs where trauma care systems can be especially fragmented resulting in worse access and outcomes for patients [60], there is scope for CBPS groups to help bridge the gaps in these much-needed services. This lack of CBPS in LMICs for people with injuries is congru-ent with the lack of reporting on the presence of CBPS groups in LMICs aside from those developed for patients who carry diagnoses like HIV/AIDS that have been given prominent status on the global health agenda. Worldwide, peer support has shown efficacy in reducing costs of care, engaging those who are often hard to reach, and providing patient-centered sup-port to empower individuals to manage and direct their care [61]. In LMICs, these effects are most often reported in groups that are focused on a defined disease process that are commonly diagnosed in LMIC settings. Twenty-six of the 53 studies in Øgård-Repål et al's review of peer

support groups for people living with HIV were undertaken in LMICs [62] and Ayala reported 27 of the 48 studies in a separate scoping review of peer and community led responses to HIV were based in the global south [63]. CBPS groups may be especially useful in providing psycho-social, knowledge, and logistical support to those affected by the high burden of injuries in LMICs in addition to ameliorating challenges in healthcare access that delay rehabilitation and medical treatment in these settings.

## Gaps in the literature

The literature is sparse & highly observational in nature, which limits the conclusions that can be drawn on the effectiveness of the groups. However, given the community-based nature of these types of support groups, it might be difficult to conduct true RCTs. This is reflected in only two RCTs fitting inclusion criteria for our review. Additional observational or survey-based studies might be the only way to discern the effects of the studies. Furthermore, many of the studies that were screened but excluded during this review involved support groups run out of healthcare facilities or under the leadership of healthcare professionals, which shows that while support groups do exist, many still are linked to health care systems.

## Limitations

Our scoping review was limited to articles in the databases we searched and to articles for which we were able to locate full texts. There were 13 (out of 112) articles not found by our search strategy that could have added additional insight. There may be more grey literature available that speaks to the existence and experience of other peer support groups, which was not covered. Furthermore, the studies included in this review were heterogeneous both in peer support group characteristics and study design, which limits the generalizability of these find-ings. Additionally, community-based peer support groups in LMICs may be underrepresented because they have not been studied or written about and thus could not be located via our search strategy. This also limits the cultural perspective of our review, as it might not include key aspects of the patient experience in regions other than North America or Europe. Finally, with such a small sample of studies, our conclusions are limited.

## Conclusions

Our study shows that community-based support groups play a role in patient recovery and emotional wellbeing after injury. There is a role for information sharing and support that may lead to increased access and retention of care for injured persons. Aside from access and reten-tion in the medical system, support groups can provide practical guidance for participants in navigating legal systems, as well as in developing connections with other patients and survivors to access to support services. Given the burden of injury that occurs in LMICs, support for development of community-based peer support groups in LMICs may increase access to care and has implications for overall improvement in healthcare delivery.

## Supporting information

**S1 Checklist. Preferred Reporting Items for Systematic reviews and Meta-Analyses exten-sion for Scoping Reviews (PRISMA-ScR) checklist.**
(DOCX)

**S1 File. Search strategies by database.**
(DOCX)

## Author Contributions

**Conceptualization:** Rashi Jhunjhunwala, Justine Davies, Kathryn Chu.

**Data curation:** Rashi Jhunjhunwala, Anusha Jayaram, Carol Mita.

**Formal analysis:** Rashi Jhunjhunwala, Anusha Jayaram.

**Funding acquisition:** Justine Davies, Kathryn Chu.

**Investigation:** Anusha Jayaram, Kathryn Chu.

**Methodology:** Carol Mita, Justine Davies, Kathryn Chu.

**Project administration:** Rashi Jhunjhunwala, Justine Davies, Kathryn Chu.

**Software:** Carol Mita.

**Supervision:** Justine Davies, Kathryn Chu.

**Writing – original draft:** Rashi Jhunjhunwala, Anusha Jayaram.

**Writing – review & editing:** Rashi Jhunjhunwala, Anusha Jayaram, Carol Mita, Justine Davies, Kathryn Chu.

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
