## [Decision Letter · Decision Letter 0]

25 Sep 2023

PONE-D-23-23344Community Support for Injured Patients: A Scoping Review and Narrative SynthesisPLOS ONE

Dear Dr. Jhunjhunwala,

Thank you for submitting your manuscript to PLOS ONE. After careful consideration, we feel that it has merit but does not fully meet PLOS ONE’s publication criteria as it currently stands. Therefore, we invite you to submit a revised version of the manuscript that addresses the points raised during the review process.

We look forward to receiving your revised manuscript.

Kind regards,

Rayan Jafnan Alharbi, Ph.D

Academic Editor

PLOS ONE

2. We notice that your supplementary file is included in the manuscript file. Please remove them and upload them with the file type 'Supporting Information'. Please ensure that each Supporting Information file has a legend listed in the manuscript after the references list.

Reviewers' comments:

Reviewer's Responses to Questions

**Comments to the Author**

1. Is the manuscript technically sound, and do the data support the conclusions?

Reviewer #1: No

Reviewer #2: Yes

Reviewer #3: Partly

2. Has the statistical analysis been performed appropriately and rigorously? 

Reviewer #1: No

Reviewer #2: Yes

Reviewer #3: N/A

3. Have the authors made all data underlying the findings in their manuscript fully available?

Reviewer #1: Yes

Reviewer #2: Yes

Reviewer #3: Yes

4. Is the manuscript presented in an intelligible fashion and written in standard English?

Reviewer #1: No

Reviewer #2: Yes

Reviewer #3: No

5. Review Comments to the Author

Reviewer #1: Although the topic of study is very important, the article is not suitable for publication.

Range of time interval of studies under review is very wide.

Table 1 could have been presented in prose, Table 2 should have been attached to the article. Poor graphs are presented, no statistical methods are used to answer the researcher's question, and it does not add new findings to the existing literature.

Reviewer #2: The studies in this review identified positive benefits of community support groups for injured people, including creation of a group identity and regaining of self-esteem, emotional support and guidance, and delivery of practical knowledge about their disease, recovery and treatment process, or ways to access medical, legal, or social support. However, there are inconsistencies in the fonts, some of which are neo-Roman. Please correct it as requested by PLOS ONE.

Reviewer #3: Overall, the paper is well-written in terms of the English language, and the conclusion aligns with the findings of this review to some extent. However, there are a few areas that need attention:

-The aim of the review appears broader than what is initially stated in the research question. Clarifying the alignment between the research question and the review's objectives would enhance the paper's coherence.

- It's noted that 13 studies were excluded due to unavailability of full texts, but there is no mention of efforts made to retrieve them, such as contacting the authors. Additionally, the potential impact of excluding these studies on the conclusions should be addressed in the limitations section.

- The discussion section requires a complete rewrite rather than just improvements. It lacks a clear flow and connection with the results section. Instead, it introduces new information that should have been integrated into the results. Furthermore, the emphasis on the scarcity of studies from low and middle-income countries diverts attention from discussing the actual results and comparing them with other studies.

I have attached a file with more detailed comments for your reference.

6. PLOS authors have the option to publish the peer review history of their article (what does this mean?). If published, this will include your full peer review and any attached files.

Reviewer #1: No

Reviewer #2: No

Reviewer #3: No

---

## [Author Response · Author response to Decision Letter 0]

27 Nov 2023

Dear Reviewers and Academic Editor, 

Thank you very much for your thorough evaluation of our manuscript, “Community Support for Injured Patients: A Scoping Review and Narrative Synthesis.” We have responded to the comments and suggestions made by all reviewers, and edited the manuscript as suggested. Please see below for our point-by-point responses. Both a marked/track changes version and unmarked version of the revised manuscript have also been submitted for review. 

Best,

Rashi Jhunjhunwala and co-authors. 

Reviewer #1

COMMENT: Although the topic of study is very important, the article is not suitable for publication. 

RESPONSE: We hope you will change this opinion after reviewing our revised manuscript.

COMMENT: Range of time interval of studies under review is very wide.

RESPONSE: We do not view this as a detriment to our study. We aimed to evaluate the research that has been done on community-based peer support groups for injured patients, and we wanted to include all research that we were able to access. We did not see a need to limit our search to any specific time frame.

COMMENT: Table 1 could have been presented in prose, Table 2 should have been attached to the article. 

RESPONSE: We chose to present table 1 in table format because we found it easier to quickly reference inclusion/exclusion criteria in this manner. Additionally, this is a format that is commonly used in scoping reviews. Table 2 is included at the bottom of the manuscript and was able to be located by other reviewers.

COMMENT: Poor graphs are presented, no statistical methods are used to answer the researcher's question, and it does not add new findings to the existing literature.

RESPONSE: The methodology utilized in this review is detailed in the Methods section of the manuscript. While no statistical methods were included, there are other valid research methodologies that exist; one of these is narrative synthesis. We refer the reviewer to the following publication from the Joanna Briggs Institute to gain insight into the ways data can be analyzed and presented in a scoping review: Pollock, Danielle1; Peters, Micah D.J.2,3,4; Khalil, Hanan5; McInerney, Patricia6; Alexander, Lyndsay7,8; Tricco, Andrea C.9,10,11; Evans, Catrin12; de Moraes, Érica Brandão13,14; Godfrey, Christina M.11; Pieper, Dawid15,16; Saran, Ashrita17,18; Stern, Cindy1; Munn, Zachary1. Recommendations for the extraction, analysis, and presentation of results in scoping reviews. JBI Evidence Synthesis 21(3):p 520-532, March 2023. | DOI: 10.11124/JBIES-22-00123

Reviewer #2

COMMENT: The studies in this review identified positive benefits of community support groups for injured people, including creation of a group identity and regaining of self-esteem, emotional support and guidance, and delivery of practical knowledge about their disease, recovery and treatment process, or ways to access medical, legal, or social support. However, there are inconsistencies in the fonts, some of which are neo-Roman. Please correct it as requested by PLOS ONE.

RESPONSE: Fonts have been standardized to adhere to PLOS ONE guidelines.

Reviewer #3

COMMENT: The aim of the review appears broader than what is initially stated in the research question. Clarifying the alignment between the research question and the review's objectives would enhance the paper's coherence.

RESPONSE: Thank you for this comment. We have revised the aims and research question. 

The aims now read: “This scoping review aims to identify the extent, distribution, benefits, utility, and impact of community-based peer support groups for injured patients”. 

The research question reads: “what research has been done on the extent, distribution, benefits, utility, and impact of health-system independent community-based support groups (i.e. those which are not based in healthcare facilities or run by healthcare professionals) for injured persons?”

COMMENT: It's noted that 13 studies were excluded due to unavailability of full texts, but there is no mention of efforts made to retrieve them, such as contacting the authors. Additionally, the potential impact of excluding these studies on the conclusions should be addressed in the limitations section. 

RESPONSE: This is a very valid point. Many of the results that we were unable to find were find were from the 1980s and early 1990s and contact information for the authors were not available. We have given more detail regarding the extent to which attempts were made to retrieve the articles in the first section of the “Results” – lines 180-183. The text now reads: 

“Full text was not recoverable for 13 studies due to unavailability online through Harvard libraries, inter-library loan, or other formal retrieval mechanisms. Four of 13 unavailable studies were published prior to 1990 and contact information for the authors was not available. Attempts to contact the other 9 authors were unsuccessful.”

We have also included the fact that these were not retrieved in the limitations, as suggested (lines 448-449): There were 13 (out of 112) articles not found by our search strategy that could have added additional insight. 

COMMENT: The discussion section requires a complete rewrite rather than just improvements. It lacks a clear flow and connection with the results section. Instead, it introduces new information that should have been integrated into the results. Furthermore, the emphasis on the scarcity of studies from low and middle-income countries diverts attention from discussing the actual results and comparing them with other studies.

RESPONSE: The discussion has been re-structured to first state main results, discussion of those results, the context in the literature with other conditions, the unique challenges of injury/trauma in formthe lack of literature in LMICs, limitations, and conclusion. The discussion on LMICs and socioeconomic element has been paired down to discuss it as an important context to injury burden, but not to take away from the main results. 

COMMENT: I have attached a file with more detailed comments for your reference. 

RESPONSE: Thank you for your thorough and thoughtful comments. We have aimed to respond to them in the track-changes document we have attached for review. These include: 

- We have standardized the language regarding “community-based peer support groups” throughout the document, and included an acronym, CBPR.

- Revised aims and research question, as stated above.

- Included more information about missing studies, as stated above.

- Included references for each component of the “study characteristics” section of the Results.

- Defined each category of “social, emotional, educational, logistical” benefits immediately prior to “figure 2”. 

- Revised/rewrote discussion, as stated above.

- Various sentences/wording have been revised as recommended.

---

## [Decision Letter · Decision Letter 1]

4 Dec 2023

PONE-D-23-23344R1Community Support for Injured Patients: A Scoping Review and Narrative SynthesisPLOS ONE

Dear Dr. Jhunjhunwala,

Thank you for submitting your manuscript to PLOS ONE. After careful consideration, we feel that it has merit but does not fully meet PLOS ONE’s publication criteria as it currently stands. Therefore, we invite you to submit a revised version of the manuscript that addresses the points raised during the review process.

We look forward to receiving your revised manuscript.

Kind regards,

Rayan Jafnan Alharbi, Ph.D

Academic Editor

PLOS ONE

Journal Requirements:

Additional Editor Comments:

Thank you for responding to the previous comments. Three minor comments were made by one of the reviewers for your consideration (lines 43, 130, and 392).

Reviewers' comments:

Reviewer's Responses to Questions

**Comments to the Author**

1. If the authors have adequately addressed your comments raised in a previous round of review and you feel that this manuscript is now acceptable for publication, you may indicate that here to bypass the “Comments to the Author” section, enter your conflict of interest statement in the “Confidential to Editor” section, and submit your "Accept" recommendation.

Reviewer #3: All comments have been addressed

2. Is the manuscript technically sound, and do the data support the conclusions?

Reviewer #3: Yes

3. Has the statistical analysis been performed appropriately and rigorously? 

Reviewer #3: N/A

4. Have the authors made all data underlying the findings in their manuscript fully available?

Reviewer #3: Yes

5. Is the manuscript presented in an intelligible fashion and written in standard English?

Reviewer #3: Yes

6. Review Comments to the Author

Reviewer #3: I have added three minor comments for the authors to consider editing. Other than these, the authors have responded well to my previous comments.

7. PLOS authors have the option to publish the peer review history of their article (what does this mean?). If published, this will include your full peer review and any attached files.

Reviewer #3: No

---

## [Author Response · Author response to Decision Letter 1]

14 Dec 2023

PONE-D-23-23344

Community Support for Injured Patients: A Scoping Review and Narrative Synthesis

Response to Reviewers

Dear Reviewers and Academic Editor, 

Thank you very much for your thorough evaluation of our manuscript, “Community Support for Injured Patients: A Scoping Review and Narrative Synthesis.” We have responded to the comments and suggestions and edited the manuscript as suggested. Please see below for our point-by-point responses. Both a marked/track changes version and unmarked version of the revised manuscript have also been submitted for review. 

Best,

Rashi Jhunjhunwala and co-authors. 

Reviewer #3

- Spacing (line 43): Extra space has been removed

- I previously commented on this issue. You need to decide whether the reference should be before or after the dot: We have standardized all references to be placed before the period. 

- Add a sentence about heterogeneity and how it could affect your findings (limitations paragraph): The following sentence was added to the Limitations paragraph: “Furthermore, the studies included in this review were heterogeneous both in peer support group characteristics and study design, which limits the generalizability of these findings.”

---

## [Editor Report · Decision Letter 2]

9 Jan 2024

Community Support for Injured Patients: A Scoping Review and Narrative Synthesis

PONE-D-23-23344R2

Dear Dr. Jhunjhunwala,

We’re pleased to inform you that your manuscript has been judged scientifically suitable for publication and will be formally accepted for publication once it meets all outstanding technical requirements.

If your institution or institutions have a press office, please notify them about your upcoming paper to help maximize its impact. If they’ll be preparing press materials, please inform our press team as soon as possible, no later than 48 hours after receiving the formal acceptance. Your manuscript will remain under strict press embargo until 2 pm Eastern Time on the date of publication. For more information, please contact onepress@plos.org.

Kind regards,

Rayan Jafnan Alharbi, Ph.D

Academic Editor

PLOS ONE

---

## [Editor Report · Acceptance letter]

23 Jan 2024

PONE-D-23-23344R2 

PLOS ONE

Dear Dr. Jhunjhunwala, 

I'm pleased to inform you that your manuscript has been deemed suitable for publication in PLOS ONE. Congratulations! Your manuscript is now being handed over to our production team.

Kind regards, 

on behalf of

Dr. Rayan Jafnan Alharbi 

Academic Editor

PLOS ONE